# SINGER: Stochastic Network Graph Evolving Operator for High Dimensional PDEs

**Mingquan Feng, Yixin Huang, Weixin Liao, Yuhong Liu, Yizhou Liu, Junchi Yan**[*]
Sch. of Computer Science & Sch. of Artificial Intelligence, Shanghai Jiao Tong University
{fengmingquan, cindyh1103, liaoweixin, yanjunchi}@sjtu.edu.cn
https://github.com/FengMingquan-sjtu/SINGER

## Abstract

We present a novel framework, StochastIc Network Graph Evolving operatoR (SINGER), for learning the evolution operator of high-dimensional partial differential equations (PDEs). The framework uses a sub-network to approximate the solution at the initial time step and stochastically evolves the sub-network parameters over time by a graph neural network to approximate the solution at later time steps. The framework is designed to inherit the desirable properties of the parametric solution operator, including graph topology, semigroup, and stability, with a theoretical guarantee. Numerical experiments on 8 evolution PDEs of 5,10,15,20-dimensions show that our method outperforms existing baselines in almost all cases (31 out of 32), and that our method generalizes well to unseen initial conditions, equation dimensions, sub-network width, and time steps.

## 1 Introduction

Partial differential equations (PDEs) are prevalent and have extensive applications in science, engineering, economics, and finance (Strauss, 2007; Folland, 2020). Given that many partial differential equations do not have analytical solutions, numerical methods like finite element methods (Huebner et al., 2001) as well as neural network based solvers (Blechschmidt & Ernst, 2021) have been established. Although these methods have been significantly advanced recently, they suffer the well-known issue of *curse of dimensionality* since the number of data points increases exponentially with respect to the dimension $d$ of independent variables (Bellman, 1966). They quickly become computationally intractable for general high dimensional problems, e.g. $d \geq 5$.

Facing the challenge, recent advances in deep-learning-based solvers propose incorporating specific PDE problem structure (Han et al., 2017; Wang et al., 2022a; Yu et al., 2018) and designing memory-efficient training strategies (Hu et al., 2024; Shang et al., 2023). These methods succeed in solving a variety of PDEs empirically. Nonetheless, they aim to solve specific instances of PDEs. Consequently, they need to retrain from scratch whenever the initial and/or boundary value changes, which is expensive and time-consuming in high-dimensional problems. To address these limitations, (Gaby et al., 2024; Gaby & Ye, 2024) obtain the solution operator of the high-dim PDEs by learning the evolution of the parameters of a sub-network over time. The training is purely based on the PDE residual loss, without requiring any spatial discretization or solutions of the PDE. Once successfully trained, the model can generalize to new initial conditions with a single inference pass. In practice, however, the training process often collapses or converges to an incorrect model.

This paper aims to address the challenge of learning high-dimensional PDE solution operators. We establish three basic assumptions of an ideal solution operator: graph topology, semigroup, and stability, and design a novel framework that inherently satisfies those assumptions with theoretical guarantees. Specifically, we propose a stochastic differential equation driven by a graph neural network (GNN) on the sub-network's network graph to model the semigroup evolution of the sub-network parameters. The GNN is designed to handle the graph topology of the sub-network, including permutation invariance and modification robustness. The stochastic noise is introduced to stabilize the evolution process and prevent divergence. We theoretically prove the semigroup evo-

---

[*]Correspondence author who is also affiliated with Shanghai Innovation Institute. This work was partly supported by NSFC (62222607, 623B1009).

lution and stability of the proposed framework and empirically demonstrate model effectiveness in accuracy, stability, and generalization on several datasets. We highlight the following **contributions**:

- Formally establish three assumptions of an ideal high-dimensional PDE solution operator: graph topology, semigroup, and stability.
- Propose **SINGER** consists of graph network, continuous evolution and stochastic noise, corresponding to the three assumptions, with theoretical and empirical justification.
- Solve eight PDEs on 5-20 dimensions by SINGER accurately and stably and generalize across initial conditions, equation dimensions, input scales, and time horizons.

## 2 RELATED WORK

Recent developments in neural PDE solvers have shown promise in overcoming the limitations of traditional numerical solvers, especially for high-dim problems. Below we discuss some of the key approaches in this area. The related work on low-dim PDEs are provided in Appendix D.

**Neural High-Dim PDEs Solvers.** For high-dim PDEs, the curse of dimensionality poses a significant challenge, thus requiring specialized methods. The existing attempts mainly fall in the physics-informed category. (Han et al., 2018; 2017) proposed the DeepBSDE solver, reforming a class of hyperbolic PDEs as backward stochastic differential equations (BSDEs) by Feynman-Kac formula. (Wang et al., 2022b;a) proposed tensor neural networks with efficient numerical integration and separable structures for solving separable PDEs such as the Schrödinger equation. The deep Ritz method (Yu et al., 2018) considers solving high-dimensional variation PDE problems by minimizing the energy functional of the PDE. However, the above techniques are limited to specific types of PDEs and may not generalize well to others. For general PDEs, (Zang et al., 2020) proposed a weak adversarial network that solves PDEs using the weak formulation. (Hu et al., 2024) designed a stochastic dimension gradient descent to reduce the computational cost.

**Neural High-Dim PDEs Operators.** The data-driven operator methods are less explored in high-dim PDEs, possibly due to the difficulty in data generating and model generalization. The only two existing works that learn the high-dim PDE solution operator are proposed by Gaby et al. (2024); Gaby & Ye (2024). The first work reduces the solution operator to the evolution of the parameters of a reduced-order model (a neural network) and introduces another network to learn the evolution in discrete time steps by minimizing PDE residual. The second work extends the first work to continuous evolution by adapting a Neural ODE (NODE) (Chen et al., 2018) framework, improving the sample efficiency. However, the above methods suffer the instability of the evolving operator, crucial for both training and inferencing. Our work follows this line of research and introduces a novel framework that inherently maintains stability, generalization, and other desirable properties.

## 3 METHODOLOGY

In the SINGER model, we utilize a neural network $U$ to approximate the solution $u(x, t)$ of the PDE. The parameters $\boldsymbol{\theta}_t$ of network $U$ evolve according to a GNN-driven ODE, where the nodes in the graph represent neurons, and the edges represent dependencies between them. The key advantage of this structure is the permutability of neurons within the same layer, which enhances the model's generalization capabilities. To combat the inherent instability in the evolution process, we introduce stochastic noise during training. This noise stabilizes the system and prevents divergence during long-term iterations. A theoretical analysis of the stochastic influence on stability is provided.

### 3.1 MOTIVATION: THREE ASSUMPTIONS

Our model design is based on three key assumptions on the sub-network $U$ and the solution operator of PDEs: the graph topology of $U$, the semigroup operator, and the stability of the solution.

The first assumption is based on the network graph $\mathcal{G}(V, E)$ of the sub-network $U$. The vertex $v_i$ denotes the $i$-th learnable parameter of $U$, and the vertex feature is the one-hot encoding of its layer index in $U$. The edge $e_{i,j}$ exists if $v_i, v_j$ are the input and output parameters, respectively, of the same neuron in $U$. We then assume such a graph is a good representation of the subnetwork $U$:

**Assumption 1** (Graph Topology)**.** *For a proper distance metric $d(\cdot, \cdot)$, there exists a constant $C > 0$ such that $d_U(U_1, U_2) \leq C d_{\mathcal{G}}(\mathcal{G}_1, \mathcal{G}_2)$ for any sub-networks $U_1, U_2$ and their network graph $\mathcal{G}_1, \mathcal{G}_2$.*

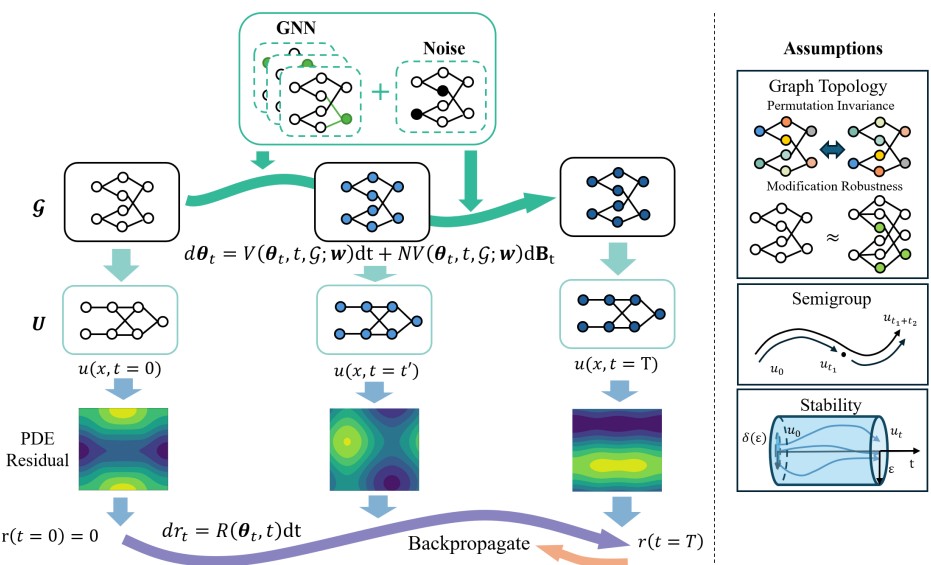

Figure 1: Left: SINGER architecture. Right: three underlying assumptions.

*This above statement leads to two properties of the sub-network $U$:*

1. *Permutation Invariance: changing the relative order of neurons of the same layer (defined as* Perm*) does not change the graph and sub-network. Formally, if $\mathcal{G}_1 := \mathrm{Perm}(\mathcal{G}_2)$, then $d_{\mathcal{G}}(\mathcal{G}_1, \mathcal{G}_2) = 0$, and $d_U(U_1, U_2) = 0$.*

2. *Modification Robustness: adding or removing nodes or edges in the graph (defined as* Modif*) does not substantially change the sub-network. Formally, for a small modification budget $\epsilon$, if $\mathcal{G}_1 := \mathrm{Modif}(\mathcal{G}_2)$ and $d_{\mathcal{G}}(\mathcal{G}_1, \mathcal{G}_2) < \epsilon$, then $d_U(U_1, U_2) < C\epsilon$.*

The above assumption holds for common neural network architecture. E.g., the permutation invariance is satisfied by fully connected layers, convolutional layers, and attention layers. Additionally, random removal of a node, e.g. via dropout, does not drastically alter the network output. Networks not following this assumption tend to be sensitive to permutation and modification noise. A desirable solution should be robust to noise, which motivates encoding this property into the model.

The second assumption is on the semigroup property of the evolving operator $T(t)$ that evolves the solution function $u$ from time 0 to time $t$. Intuitively, the semigroup property states that the operator $T$ is consistent over time, i.e. the evolution from time 0 to $t$ is the same as the concatenation of evolutions from 0 to any middle time $t_i$ and from time $t_i$ to $t$. The formulation is:

**Assumption 2** (Semigroup). *Suppose $T$ is a mapping $[0, +\infty) \to \mathscr{L}(X)$, where $X$ is the space of the PDE solution function $u$ and $\mathscr{L}(X)$ denotes the set of bounded operators on $X$. The mapping $T$ satisfies the semigroup property, i.e.*

$$T(t_1)T(t_2) = T(t_1 + t_2), \quad \text{for any } t_1, t_2 \geq 0 \tag{1}$$

$$T(0) = \boldsymbol{I}, \quad \text{where } \boldsymbol{I} \text{ is the identity operator.} \tag{2}$$

The above assumption holds for initial value problems of PDEs (Farkas et al., 2011). This property arises from the physical interpretation of PDEs, where the time evolution of the solution is independent. Similar to the Markov property, the future state depends only on the current state, with no memory of the past. Encoding this prior can prevent it from learning non-physical solutions.

The third assumption is the stability of the solution function $u$, defined as the boundedness of the evolution of $u$ under a small perturbation of initial condition with formulation as follows.

**Assumption 3** (Stability). *The solution function $u$ of the PDE is stable under small perturbation of the initial condition, i.e. $\forall \varepsilon > 0$, there exists a $\delta > 0$ such that for any initial condition $u_0$ and $u_0'$ with $\|u_0 - u_0'\| < \delta$, the solution functions satisfy $\|T(t)(u_0) - T(t)(u_0')\| < \epsilon$ for any $t \geq 0$.*

Table 1: Comparison on assumption satisfaction of neural operators.

| Method | Graph Topology | Semigroup | Stability |
|---|---|---|---|
| NODE (Gaby & Ye, 2024) | No | Yes | No |
| PINO (Li et al., 2024) | No | No | Yes |
| **SINGER (ours)** | **Yes** | **Yes** | **Yes** |

The stability holds for many basic PDEs, such as heat equation, reaction-diffusion equation, and wave equation(Shirinabadi & Talebi, 2011). The stability is rooted from the *well-posedness* of PDEs (Hilditch, 2013). A PDE is well-posed if its solution exists, is unique, and continuously depends on the initial and boundary conditions. The stability we define refers to the continuous dependence of the PDE solution on the initial condition. Stability is of paramount importance in both theoretical analysis and numerical solutions of PDEs. For example, in our experiments, models without stability would collapse during training (see Table 2).

As compared in Table 1, the three assumptions are non-trivial for a neural network operator. Consider two baselines NODE (Gaby & Ye, 2024) and PINO(Li et al., 2024)(modified). The NODE model flatten the initial sub-network parameters into a vector and evolve the it by a freely learnable neural ODE. The flattening operation violates the graph topology assumption, and a free NODE model is not guaranteed to be stable. The semigroup property is satisfied by the integration operator in neural ode forwarding.

The PINO learns a network mapping the input data and time $t$ directly to the solution function values at sampling points and time $t$. It is not directly applicable to high-dim problems, so we modify it to mapping the initial sub-network parameters $\theta_0$ and time $t$ to the solution sub-network parameters $\theta_t$. The PINO violates the graph topology if the $\theta$ is flattened. And even if the graph is preserved, the semigroup property is not guaranteed, since the model is a black-box function with regard to the time $t$. Whereas the stability is generally guaranteed due to the continuity of network.

The violation of the assumption usually results in poor experiment performance or even collapse of the model, as we will show in the experiment section. Our model, SINGER, is designed to satisfy all three assumptions, as introduced and justified in the following subsections.

## 3.2 FORMULATION AND ALGORITHM

We follow the problem setting in Gaby et al. (2024) and consider the initial value problem of the evolution PDE defined as follows:

$$\begin{cases} \partial_t u(x,t) = F[u](x,t), & x \in \Omega, t \in (0, T_e], \\ u(x,0) = u_0(x), & x \in \Omega, \end{cases} \quad (3)$$

where $\Omega$ is an open bounded set in $\mathbb{R}^d$, $F$ is a possibly nonlinear differential operator of functions $u : \Omega \times (0, T_e] \to \mathbb{R}$, $u_0 : \Omega \to \mathbb{R}$ stands for an initial value, and $T_e > 0$ is terminal time. The solution is defined in the strong sense, thus $u$ is assumed continuous: $u \in C^{2,1}(\bar{\Omega} \times (0, T_e))$

The SINGER parameterises solution $u(\cdot, t)$ by a sub-network $U_{\boldsymbol{\theta_t}}$, with network graph $\mathcal{G}$ and parameters $\boldsymbol{\theta_t}$ at time $t$. The parameter evolution is modeled by an operator-network $\boldsymbol{V}$ with parameters $\boldsymbol{w}$. The evolution of the parameters is given by the following equation:

$$\mathrm{d}\boldsymbol{\theta_t} = \boldsymbol{V}(\boldsymbol{\theta_t}, t, \mathcal{G}; \boldsymbol{w})(\mathrm{d}t + \boldsymbol{N}\,\mathrm{d}\mathbf{B}_t), \quad (4)$$

where $\boldsymbol{N}$ is a noise matrix, and $\mathbf{B}_t$ is a Brownian motion. The operator-network is implemented as a GNN to handle the graph topology of sub-network $U$. The stochastic noise term $\boldsymbol{N}\,\mathrm{d}\mathbf{B}_t$ is introduced to stabilize the evolution process and prevent divergence.

The network $\boldsymbol{V}$ instantiate the Message Passing Neural Network (MPNN) (Gilmer et al., 2017), a general framework of GNN, to handle the graph topology of sub-network $U$. A $D$-layer MPNN consists of $D$ pairs of message and update functions, denoted as $M_i$ and $P_i$ respectively:

$$m_v^{i+1} = \sum_{w \in N(v)} M_t\left(h_v^i, h_w^i\right), \qquad h_v^{i+1} = P_t\left(h_v^i, m_v^{i+1}\right), \quad (5)$$

---

**Algorithm 1** Training pipeline for SINGER

---
**Input:** parameter set $\vartheta$, sub-network structure $U$, network graph $\mathcal{G}$ and operator-network $V$
**Output:** optimal weights $w$
 1: Sample $\{\boldsymbol{\theta}_0^k\}_{k=1}^K$ uniformly from the parameter set $\vartheta$.
 2: Initialize Eq.4 and Eq.6 with $\boldsymbol{\Theta}_0^k = [\boldsymbol{\theta}_0^k, 0]$
 3: Calculate loss $\hat{l}(\boldsymbol{w}) = \frac{1}{K} \sum_{k=1}^K \left[ r_T(\boldsymbol{\theta_0^k}) \right]$ by integrating Eq.4 and Eq.6.
 4: Use adjoint method according to Eq.7 to calculate $\nabla_{\boldsymbol{w}} \hat{l}(\boldsymbol{w})$. Minimize the loss w.r.t $\boldsymbol{w}$

---

**Algorithm 2** Testing pipeline for SINGER

---
**Input:** initial condition $u_0$, sub-network $U$, network graph $\mathcal{G}$ and trained operator-network $V$
**Output:** PDE solution $u(\cdot, t)$
 1: Estimate $\boldsymbol{\theta_0}$ using gradient descent with Bernoulli noise: $\boldsymbol{\theta_0} = \arg\min_{\boldsymbol{\theta_0}} \|u_0 - U_{\boldsymbol{\theta_0}}\|_2$.
 2: Initialize Eq.4 with $\boldsymbol{\Theta}_0 = \boldsymbol{\theta}_0$
 3: Integrate Eq.4 to obtain network parameters $\boldsymbol{\theta}_t$.
 4: Place $\boldsymbol{\theta}_t$ into sub-network U and retrieve PDE solution $u(\cdot, t) = U_{\boldsymbol{\theta}_t}$

---

where $h_v^i$ is the hidden state of node $v$ at layer $i$, $N(v)$ is the set of neighbors of node $v$. The initial and final hidden state $h_v^1, h_v^D$ is the input and output of MPNN, respectively. In our model, the input $h_v^1$ is the node feature, i.e., the one-hot vector of the parameter's layer index. The output $h_v^D$ is of the same shape as the input and is decoded by reading out the element at the layer index.

In order to optimize the parameters of the operator-network $V$, we augment the above equation with accumulated residual $r_t$ in strong sense:

$$\mathrm{d}r_t = R(\boldsymbol{\theta_t}, t)\,\mathrm{d}t := \left\| \boldsymbol{\nabla_\theta} U_{\boldsymbol{\theta_t}} \cdot \mathrm{d}\boldsymbol{\theta_t} - F[U_{\boldsymbol{\theta_t}}]\,\mathrm{d}t \right\|^2, \tag{6}$$

where $F$ is the right-hand side differential operator of the PDE, and $U_{\boldsymbol{\theta_t}}$ is the solution of the PDE at time $t$ with parameters $\boldsymbol{\theta_t}$. The initial residual $r_0$ is set to zero.

The training optimization problem is defined as finding the optimal parameters $w$ that minimize the final residual $r_T$ over distribution of initial parameters $\boldsymbol{\theta_0}$: $\boldsymbol{w}^* = \arg\min_{\boldsymbol{w}} \mathbb{E}_{\boldsymbol{\theta_0}}[r_T]$. To optimize the parameters and avoid high memory costs, we adopt the scheme presented by Liu et al. (2019). L is used here to represent the loss function $\mathbb{E}_{\boldsymbol{\theta_0}}[r_T]$. The gradient is: $\frac{\partial \hat{L}}{\partial \boldsymbol{w}} = \frac{\partial L}{\partial r_T} \frac{\partial r_T}{\partial \boldsymbol{w}}$. If we denote $\frac{\partial r_t}{\partial \boldsymbol{w}}$ as $\alpha_t$ and $\frac{\partial \boldsymbol{\theta}_t}{\partial \boldsymbol{w}}$ as $\boldsymbol{\beta_t}$, we can get the following relationship:

$$
\begin{aligned}
\mathrm{d}\alpha_t &= \frac{\partial R(\boldsymbol{\theta_t}, \boldsymbol{t})}{\partial \boldsymbol{\theta}_t} \boldsymbol{\beta_t}\,\mathrm{d}t \\
\mathrm{d}\boldsymbol{\beta_t} &= \left( \frac{\partial \boldsymbol{V}}{\partial \boldsymbol{w}} + \frac{\partial \boldsymbol{V}}{\partial \boldsymbol{\theta}_t} \boldsymbol{\beta}_t \right)\mathrm{d}t + \boldsymbol{N}\left( \frac{\partial \boldsymbol{V}}{\partial \boldsymbol{w}} + \frac{\partial \boldsymbol{V}}{\partial \boldsymbol{\theta}_t} \boldsymbol{\beta}_t \right)\mathrm{d}\boldsymbol{B}_t
\end{aligned}
\tag{7}
$$

In a way similar to the adjoint method in Neural ODE, Eq.7 is solved alongside the original SDE system. This approach eliminates the need to store intermediate states, effectively reducing the original memory burden. In the training phase, SINGER seeks to find the optimal operator network using the adjoint method given in Eq.7. The algorithm is described in detail in Algorithm 1.

Given a trained operator network, we can solve high-dimensional PDEs in the following steps. Firstly, given initial value $u_0$, we find a $\boldsymbol{\theta_0} = \arg\min_{\boldsymbol{\theta_0}} \|u_0 - U_{\boldsymbol{\theta_0}}\|_2$. Secondly, we solve Eq.4 to retrieve $\boldsymbol{\theta}_t$. Finally, the PDE solution is generated using $\boldsymbol{\theta}_t$ and sub-network $U$. The detailed algorithm is provided in Algorithm 2.

## 3.3 THEORETICAL ANALYSIS

### 3.3.1 GRAPH TOPOLOGY AND SEMIGROUP PROPERTY OF SINGER

Firstly we will focus on the Graph Topology assumption. Since all our operations in the SINGER preserve permutation invariance (element-wise operation and MPNN), the model itself preserves permutation invariance. As for Modification Robustness, the GNN holds continuity to graph structure (Han et al., 2024), which means that when edges are added or removed, GNN's output won't drastically change. Thus, our model observes Assumption 1.

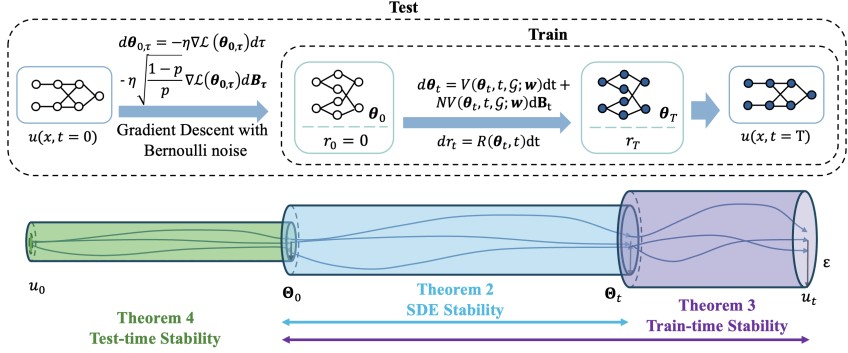

Figure 2: SIGNER's training/testing and the algorithmic process considered in each theorem.

Then we shall discuss Assumption 2 on our model. As the solution operator is a stochastic process, the definition of semigroup, especially for equality, needs an extension to the stochastic version.

**Definition 1** (Stochastic Semigroup). *Suppose $T = \{T_t(\omega), t \in [0, +\infty)\}$ is a stochastic process, where $\omega$ being Brownian path, the stochastic variable $T_t(\omega) \in \mathscr{L}(X)$ is solution operator, $X$ is the space of the solution function $u$ and $\mathscr{L}(X)$ denotes the set of bounded operators on $X$. Denote almost surely by a.s., denote identically distributed by $\sim$. Then $T$ is:*

1. *Strong Semigroup, if for $\forall t_1, t_2 \geq 0, u \in X$:*

$$T_{t_1+t_2}(\omega)(u) = T_{t_2}T_{t_1}(\omega)(u) \ a.s., \quad T_0(\omega)(u) = u \ a.s., \tag{8}$$

2. *Weak Semigroup, if for $\forall t_1, t_2 \geq 0, u \in X$:*

$$T_{t_1+t_2}(\omega)(u) \sim T_{t_2}T_{t_1}(\omega)(u), \quad T_0(\omega)(u) \sim u, \tag{9}$$

The SINGER evolution operator is a stochastic semigroup with a theoretical guarantee under mild conditions (all proofs provided in the Appendix B).

**Theorem 1.** *If $\mathbf{V}(\boldsymbol{\theta}_t, t, \mathcal{G}; \boldsymbol{w})$ is square-integrable w.r.t. time $t$ and sub-network $U_{\boldsymbol{\theta}_t}$ is twice continuously differentiable w.r.t. parameter $\boldsymbol{\theta}_t$, then the solution operator $T$ of $U_{\boldsymbol{\theta}_t}$ satisfies both strong and weak semigroup property.*

### 3.3.2 STABILITY OF SINGER

We will theoretically analyze the stability of the proposed SINGER. The overall algorithm considered in this section is depicted in Fig. 2.

To examine the stability of an SDE, we initialize the system at two slightly different value $(\boldsymbol{\theta_0}, r_0)$ and $(\boldsymbol{\theta_0^e}, r_0^e) = (\boldsymbol{\theta_0} + \boldsymbol{\varepsilon_0^\theta}, r_0 + \varepsilon_0^r)$, where $(\boldsymbol{\varepsilon_0^\theta}, \varepsilon_0^r)$ is the initial perturbation. The stability is assessed based on the long-term behavior of the perturbed model, i.e., how $(\boldsymbol{\varepsilon_t^\theta}, \varepsilon_t^r)$ evolve in the long run.

$$\begin{aligned} \mathrm{d}\boldsymbol{\varepsilon_t^\theta} &= [\boldsymbol{V}(\boldsymbol{\theta_t^e}, t, \mathcal{G}; \boldsymbol{w}) - \boldsymbol{V}(\boldsymbol{\theta_t}, t, \mathcal{G}; \boldsymbol{w})]\,\mathrm{d}t + \boldsymbol{N}[\boldsymbol{V}(\boldsymbol{\theta_t^e}, t, \mathcal{G}; \boldsymbol{w}) - \boldsymbol{V}(\boldsymbol{\theta_t}, t, \mathcal{G}; \boldsymbol{w})]\,\mathrm{d}\mathbf{B}_t \\ &= \boldsymbol{V_\Delta}(\boldsymbol{\varepsilon_t^\theta}, t, \mathcal{G}; \boldsymbol{w})\,\mathrm{d}t + \boldsymbol{N}\boldsymbol{V_\Delta}(\boldsymbol{\varepsilon_t^\theta}, t, \mathcal{G}; \boldsymbol{w})\,\mathrm{d}\mathbf{B}_t \end{aligned} \tag{10}$$

$$\mathrm{d}r_t = [R(\boldsymbol{\theta_t^e}, t) - R(\boldsymbol{\theta_t}, t)]\,\mathrm{d}t = R_\Delta(\boldsymbol{\varepsilon_t^r}, t)\,\mathrm{d}t\,x \tag{11}$$

Here we assume that the Brownian motions have the same sample path for both of the initializations to do subtraction. Given the above description of the problem, we can now formally define stability, which is adapted from Liu et al. (2019):

**Definition 2.** *(Lyapunov stability of SDE). The solution $\boldsymbol{\varepsilon_t} = 0$ is almost surely exponentially stable if $\limsup_{t \to \infty} \frac{1}{t} \log \|\boldsymbol{\varepsilon_t}\| < 0$ a.s. $\forall \boldsymbol{\varepsilon_0} \in \mathbb{R}^n$.*

Liu et al. (2019) has already proved the stability of Eq.10-like SDE in the following theorem:

**Theorem 2.** *For Eq.10-like SDE, if $V(\boldsymbol{\theta_t}, t, \mathcal{G}; \boldsymbol{w})$ is L-Lipschitz continuous w.r.t. $\boldsymbol{\theta_t}$, then it has a unique solution with the property $\limsup_{t\to\infty} \frac{1}{t} \log \|\boldsymbol{\varepsilon_t}\| \leq -(\frac{N^2}{2} - L)$ a.s. for any $\boldsymbol{\varepsilon_0} \in \mathbb{R}^n$. In particular, if $N^2 > 2L$, the solution $\boldsymbol{\varepsilon_t} = 0$ is a.s. exponentially stable.*

However, we can not apply Theorem 2 directly to Eq.11 since there is no diffusion term in it. It is worth noting that Eq.11 is non-autoregressive, i.e. $\frac{\mathrm{d}r_t}{\mathrm{d}t}$ is not influenced by $r_t$. Any perturbation to the initial value of $r_0$ will result in only a constant-level disturbance at time $t$. Given certain assumptions on the system and results from Liu et al. (2019), we can prove the following theorem:

**Theorem 3.** *If $V(\boldsymbol{\theta_t}, t, \mathcal{G}; \boldsymbol{w})$ and $R(\theta_t, t)$ is L-Lipschitz continuous w.r.t. $\theta_t$ and $N^2 > 2L$, then the solution $(\boldsymbol{\varepsilon_t^\theta}, \varepsilon_t^r) = \mathbf{0}$ is almost surely exponentially stable.*

Finally, we shall examine the SINGER stability during training. Adapted from Assumptions 3, the definition of its train-time stability is formulated as follows:

**Definition 3.** *(SINGER train-time stability)* $\forall \varepsilon > 0, \exists \delta$, *if* $\|\boldsymbol{\Theta_0} - \boldsymbol{\Theta_0^e}\| < \delta$, *then* $\|U_{\boldsymbol{\theta_t^e}} - U_{\boldsymbol{\theta_t}}\| < \varepsilon$.

where $(\boldsymbol{\theta_t}, r_t)$ is denoted as $\boldsymbol{\Theta_t}$ for simplicity. Given Theorem 3 and some assumptions on neural networks, we can prove the train-time stability of the SINGERe.

**Theorem 4.** *If the $\boldsymbol{\Theta_t}$ evolution process satisfies all conditions in Theorem 3, then SINGER is stable during training.*

We further focus on the SINGER stability during testing. In the test phase, the PDE initial condition $u_0$ is provided as input. The gradient descent method is applied to seek initial sub-network parameters $\boldsymbol{\theta_0} = \arg\min \mathcal{L}(u_0, U_{\boldsymbol{\theta_0}})$, where $\mathcal{L}$ is the loss function. We first define the test stability:

**Definition 4.** *(SINGER test-time stability)* $\forall \varepsilon > 0, \exists \delta$, *if* $\|U_{\boldsymbol{\theta_0}} - U_{\boldsymbol{\theta_0^e}}\| < \delta$, *then* $\|U_{\boldsymbol{\theta_t^e}} - U_{\boldsymbol{\theta_t}}\| < \varepsilon$.

To ensure the stability of the process of seeking $\boldsymbol{\theta_0}$, we use a gradient descent algorithm with Bernoulli noise. Following a similar proof scheme of Theorem 3 and Theorem 4, we can further confirm SINGER test-time stability.

**Theorem 5.** *If both the $\boldsymbol{\theta_0}$ fitting process and the $\boldsymbol{\Theta_t}$ evolution process satisfies all conditions in Theorem 3, then SINGER is stable during testing.*

## 4  EXPERIMENTS

We validate the proposed architecture on 8 benchmark high-dim PDEs and compare it with SOTA methods. The experiments focus on accuracy, generalization, stability, and assumption fulfillment. Our model is tested on 5- to 10-dimensional PDEs and evaluated for its ability to generalize across different initial conditions, dimensions of the PDE, the width of the sub-network, and time steps.

We compare our method with 4 baselines. The first one is the non-noise version of our method, denoted as SINGER(-N). We also choose NODE(Gaby & Ye, 2024) and PINOLi et al. (2024) as baselines due to their closeness to our model. The original PINO learns the mapping of function values, which is not suitable for high-dimensional PDEs. Therefore, we modified it to learn the evolution of sub-network parameters. After modification, PINO can be regarded as a variant of NODE without semigroup property. The last baseline is the NODE with noise, denoted as NODE(+N), regarded as the non-graph version of SINGER.

### 4.1  IMPLEMENTATION DETAILS

In all experiments, we implement the SINGER with a 3-layer message passing GNN, with the message and update function being 4-layer residual networks (ResNets). The hidden dimension is 100, and the noise type is Bernoulli (i.e. dropout) with a probability of 0.1. The validity of using Bernuolli noise is justified in Appx A. We also implement a NODE baseline using a ResNet with 1000 hidden dimensions. The ODE is integrated with the Runge-Kutta (Butcher, 1996) with a time step $1/10$ of the time horizon. The PINO baseline is also with 1000 hidden dimensions. It is implemented with the trick from the Consistency Model (Song et al., 2023), i.e., embedding the time index by an encoder network, then adding the encoded feature to other inputs and feeding into ResNet. The training set consists of 100000 randomly generated $\theta$ from the normal distribution, and the test set is 100 randomly generated $\theta$ from another normal distribution with slightly different mean and variance, as suggested by Gaby & Ye (2024). The training is done with Adam optimizer with a learning

Table 2: Relative error on PDEs with explicit solution, the **best** and second best are marked, and the * symbol denotes collapse in training.

| Eqn | Heat | | | | HJB | | | |
|---|---|---|---|---|---|---|---|---|
| Dim | d=5 | d=10 | d=15 | d=20 | d=5 | d=10 | d=15 | d=20 |
| SINGER | **0.0045** | **0.0041** | **0.0046** | **0.0175*** | **0.0036** | 0.0027 | **0.0099** | 0.0052 |
| SINGER(-$N$) | 0.0137* | 0.0294* | 0.0371* | 0.0507 | 0.0037 | 0.0027 | 0.0106 | **0.0047** |
| NODE | 0.0194 | 0.0342* | 0.0502* | 0.0957* | 0.0042 | 0.0090 | 0.0280 | 0.0602 |
| NODE(+$N$) | 0.0213 | 0.0220 | 0.0196 | 0.0751 | 0.0040 | 0.0041 | 0.0197 | 0.0398 |
| PI-NO | 0.0543 | 0.0229 | 0.0296 | 0.0567 | 0.1075 | 0.5387 | 0.9386 | 3.5078 |

Table 3: Relative error on various PDEs, the **best** is marked, and the * denotes collapse in training.

| Eqn | Pricing | | | | Sine-Gorden | | | |
|---|---|---|---|---|---|---|---|---|
| Dim | d=5 | d=10 | d=15 | d=20 | d=5 | d=10 | d=15 | d=20 |
| SINGER | **0.0510** | **0.0217** | **0.0308** | **0.0519** | **0.00051** | **0.00054** | **0.00073** | **0.00066** |
| NODE(+$N$) | 0.0541 | 0.0258 | 0.0473 | 0.0689 | 0.05431 | 0.00065 | 0.00074 | 0.00085 |
| Eqn | Burgers | | | | Reaction Diffusion | | | |
| SINGER | **0.0020** | **0.0032** | **0.0079** | **0.0088** | 0.0022 | **0.0016** | **0.0018** | **0.0020** |
| NODE(+$N$) | 0.0274* | 0.0068 | 0.0080 | 0.1729 | **0.0021** | 0.0024 | 0.0245 | 0.0056 |
| Eqn | HJBLQ | | | | Allen-Cahn | | | |
| SINGER | **0.0041** | **0.0033** | **0.0035** | **0.0037** | **0.00052** | **0.00071** | **0.00088** | **0.00077** |
| NODE(+$N$) | 0.0070 | 0.0039 | 0.0041 | 0.0059 | 1.16401* | 0.13258 | 0.00125 | 0.00373 |

rate of 0.0005 and the batch size is 64. The metric used in experiments is the $L_2$ relative error (L2RE) defined as: $\text{L2RE}(u, u') = \frac{\|u - u'\|_2}{\|u\|_2}$. The function norm is evaluated by the Monte Carlo method with 1000 samples. Unless otherwise stated, the L2RE in the main text and tables denotes $\text{L2RE}(u_{ref}, u_{pred})$, where $u_{ref}$ is the reference solution and $u_{pred}$ is the predicted solution.

We choose two PDEs with explicit solutions, the Heat equation and HJB equation (Evans, 2022):

$$\partial_t u = \Delta_x u, \quad u(x,t) = \int g(y) N(y - x, 2t I_d) dy, \quad x \in [-1, 1]^d, t \in [0, 0.1] \tag{12}$$

$$\partial_t u = \Delta_x u - \frac{1}{2}|\nabla_x u|^2, \quad u(x,t) = -2\ln \int \frac{e^{-|x-y|^2/(4(1-t))-g(y)/2}}{(4\pi(1-t))^{d/2}} dy, \quad x \in \mathbb{R}^d, t \in [0, 1] \tag{13}$$

where $N$ stands for the density of Gaussian, $g(y)$ is the initial function, and the HJB is time-reversed for consistency with the experiment. The subnetwork architecture for heat and HJB are two-layer networks, reflecting the structure of the solution:

$$u_\theta(x) = \sum_{i=1}^{w} c_i \tanh\left(a_i^\top \sin(\pi(x - \beta)) - b_i\right), \theta = (\beta, a, b, c) \in \mathbb{R}^d \times \mathbb{R}^{d \times w} \times \mathbb{R}^w \times \mathbb{R}^w \tag{14}$$

$$u_\theta(x) = \sum_{i=1}^{w} w_i e^{-|a_i \odot (x - b_i)|^2/2}, \quad \theta = (a, b, w) \in \mathbb{R}^{d \times w} \times \mathbb{R}^{d \times w} \times \mathbb{R}^w \tag{15}$$

where $d$ is input dimension and $w$ is sub-network width. See remaining six PDEs in Appendix C

## 4.2 RESULTS AND DISCUSSION

The experiments are organized to answer the following research questions:

**RQ1**: How does the SINGER accurately and stably solve high-dimensional PDEs?

**RQ2**: How does the SINGER generalize to unseen longer time indexes and higher dimensions?

**RQ3**: How does the SINGER satisfy the three assumptions in Section 3.1?

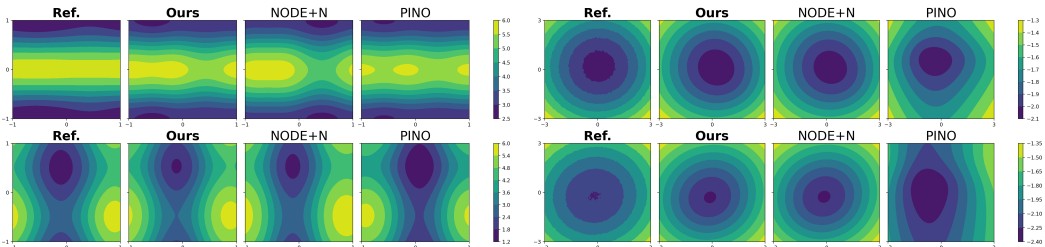

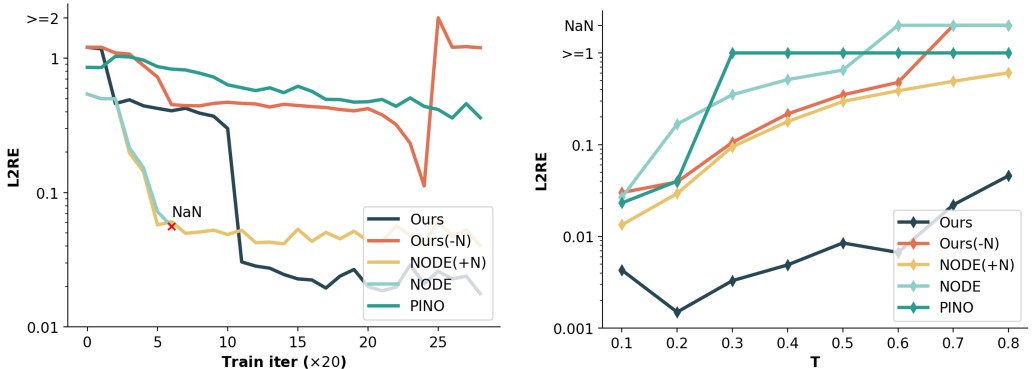

Figure 3: Examples on 10-dim Heat(left) and HJB(right). Only the first two spacial dim are shown.

Figure 4: Models trained on 10-dimensional heat equation with $t = 0.1$. **(Left)** Relative error in the early stage (560 iterations) of training. **(Right)** time extrapolation to $t = 0.8$.

**RQ1: Accuracy and Stability** We first evaluate the accuracy and stability of SINGER on PDEs with explicit solutions, i.e. the heat and HJB equations, across dimensions from 5 to 20. The results are shown in Table 2. Our model outperforms almost all baselines w.r.t. relative error. Adding noise into the training is beneficial for both our model and NODE, effectively improving accuracy and preventing the collapse of training in the heat equation. The PINO model performs poorly in all cases, with a relative error of over 0.05 in most cases. Examples of the solution are shown in Fig. 3, where the proposed model accurately captures the solution of the heat equation and HJB equation.

We select a strong baseline NODE(+N) as the comparison baseline for the 6 more PDEs. Those PDEs have no explicit solution in general, and the reference solution is generated by the DeepBSDE (Han et al., 2018). The generation details are in Appendix C. As illustrated in Table 3, the SINGER outperforms NODE(+N) with a clear margin in almost all cases. SINGER is also more stable than NODE(+N), which collapses in training in the Burgers and Reaction-Diffusion equations.

To further investigate the performance of the models, we visualize the early stage of training in Fig. 4(left). The NODE model collapses in the first few epochs, while the proposed model and NODE(+N) are more stable during training. The proposed model converges faster than NODE(+N) in the longer term, which is consistent with the results in Table 2 and Table 3. The PINO is stable but converges slower than the proposed model and NODE(+N). The results above demonstrate that the noise is crucial for training stability, and the graph topology and semigroup property can accelerate the convergence of the model in the early stage, possibly due to the incorporation of the prior knowledge, saving the model from exploring the wrong direction.

**RQ2: Generalization** We test the zero-shot generalization for unseen time periods, subnetwork widths, and even higher dimensions. The results are shown in Fig. 4(right) and Table 4. Our model outperforms all baselines in all cases, showing its superior generalization ability. In Fig. 4(right), the models are trained on a short time horizon of $t = 0.1$ and tested on a long horizon of $t = 0.8$. The proposed model maintains a low error as the horizon extends, whereas other models experience a substantial increase in error or break down into NaN values. Notably, removing the noise from our model will lead to a collapse in training, highlighting the importance of noise in training.

Table 4: Zero-shot generalization of models trained in 10-dim and 80-width to higher PDE dimensions and larger subnetwork width.

| Dim | 10 | 11 | 12 | 13 | 14 | 15 |
|---|---|---|---|---|---|---|
| **Ours** | **0.0041** | **0.0159** | **0.0255** | **0.0448** | **0.0672** | **0.1194** |
| NODE+N+I | 0.0220 | 0.0789 | 0.1492 | 0.1259 | 0.1433 | 0.1646 |
| Width | 80 | 85 | 90 | 95 | 100 | 105 |
| **Ours** | **0.0041** | **0.0045** | **0.0061** | **0.0049** | **0.0051** | **0.0045** |
| NODE+N+I | 0.0220 | 0.0531 | 0.1079 | 0.1306 | 0.1404 | 0.1857 |

Table 5: Ablation study on the violation(residual) of three assumptions.

| | Ours | | | Ablation | |
|---|---|---|---|---|---|
| Assumption | Before Train | After Train | Model | Before Train | After Train |
| Graph Topology | $1.6 \times 10^{-14}$ | $1.1 \times 10^{-14}$ | NODE(+N) | 0.00205 | 0.00478 |
| Semigroup | $1.7 \times 10^{-13}$ | $4.7 \times 10^{-10}$ | PINO | 0.14441 | 0.01484 |
| Stability | 0.29058 | 0.00057 | SINGER(-N) | 0.29199 | 0.02538 |

In Table 4, we add neurons to the subnetwork in the test time. Since generally NODE can not handle modification of subnetwork, the proposed model is compared with NODE(+N+I), where the +I means ignoring the additional neurons in the subnetwork. The models are trained in 10-dim and 80-width and can generalize to higher problem dimensions and wider subnetworks without finetuning. The proposed model outperforms NODE(+N+I) in all cases, showcasing its generalization ability. As dimension increases, SINGER error increases slower than NODE(+N+I). When subnetwork width increases, SINGER keeps error constantly while NODE(+N+I) increases error significantly. The generalization arises from the preservation of graph topology since the modification of subnetwork is essentially a modification of graph structure. The generalization of problem dimensions is a novel metric in the existing high-dimension PDE solver literature, and this ability might be useful in a pre-trained foundation model of PDE solver such as PDEformer (Ye et al., 2024).

**RQ3: Assumption Satisfaction** Lastly, we numerically check the assumptions in Section 3.1. To quantify the violation of assumptions we define the following metrics:

**1) Graph Topology**: $\mathrm{L2RE}(T(t)(u), T(t)(u_{\mathrm{perm}}))$, where $u_{\mathrm{perm}}$ is the randomly neuron-permuted input of $u$.

**2) Semigroup**: $\mathrm{L2RE}(T(t)(u), T(t - t_i)T(t_i)(u))$, where $t_i$ is a randomly selected time index.

**3) Stability**: $\mathrm{L2RE}(T(t)(u), T(t)(u_{\mathrm{noise}}))$, where $u_{\mathrm{noise}}$ is the input of $u$ with added noise.

Table 5 shows the metrics of the proposed SINGER and its ablation models with respect to the assumptions, before and after training. For the graph topology, the SINGER inherently satisfies the assumption by its graph network module, while the ablation model does not and even can not learn from training. For the semigroup, the SINGER also inherently satisfies the assumption by its ode framework, while the ablation model does not and learns from training with limited improvement. For stability, both the SINGER and ablation models are not satisfied initially, but the SINGER improves significantly after training, while the ablation model improves a little. The results demonstrate that the proposed model automatically satisfies the first two assumptions and can learn from training to improve the stability property. Whereas the ablation model merely learns from training and generally can not satisfy the assumptions after training.

## 5 CONCLUSION

We have introduced a novel architecture for learning high-dimensional PDEs solution operators based on a graph neural network. Our model is designed to satisfy three key assumptions: graph topology, semigroup property, and stability. We demonstrate the effectiveness of our approach through extensive experiments. Future work and broader impact are discussed in Appendix E.

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

## A  APPROXIMATION OF DROPOUT

In this section, we form the dropout process as by Liu et al. (2019) to analyze its stability performance using Itô process. The dropout layer can be modeled using a Bernoulli process:

$$
\begin{aligned}
\boldsymbol{\theta_{n+1}} &= \boldsymbol{\theta_n} + \boldsymbol{V}(\boldsymbol{\theta_n}, n, \mathcal{G}; \boldsymbol{w}) \odot \frac{\boldsymbol{\gamma_n}}{p} \\
&= \boldsymbol{\theta_n} + \boldsymbol{V}(\boldsymbol{\theta_n}, n, \mathcal{G}; \boldsymbol{w}) + \boldsymbol{V}(\boldsymbol{\theta_n}, n, \mathcal{G}; \boldsymbol{w}) \odot (\frac{\boldsymbol{\gamma_n}}{p} - \mathbf{I}) \\
&= \boldsymbol{\theta_n} + \boldsymbol{V}(\boldsymbol{\theta_n}, n, \mathcal{G}; \boldsymbol{w}) + \boldsymbol{V}(\boldsymbol{\theta_n}, n, \mathcal{G}; \boldsymbol{w}) \odot [\sqrt{\frac{1-p}{p}}(\sqrt{\frac{p}{1-p}}(\frac{\boldsymbol{\gamma_n}}{p} - \mathbf{I}))] \\
&= \boldsymbol{\theta_n} + \boldsymbol{V}(\boldsymbol{\theta_n}, n, \mathcal{G}; \boldsymbol{w}) + [\sqrt{\frac{1-p}{p}} \boldsymbol{V}(\boldsymbol{\theta_n}, n, \mathcal{G}; \boldsymbol{w})] \odot \boldsymbol{z_n}
\end{aligned}
\tag{16}
$$

where $\boldsymbol{\gamma_n} \sim \mathcal{B}(1, p)$ and $\odot$ represents the Hadamard product. The random variable $\boldsymbol{z_n}$ is formulated to approximate the standard Gaussian distribution.

Given the Itô process $\boldsymbol{\theta_t}$ satisfying $\mathrm{d}\boldsymbol{\theta_t} = \boldsymbol{V}(\boldsymbol{\theta_t}, t, \mathcal{G}; \boldsymbol{w}) \, \mathrm{d}t + \sqrt{\frac{1-p}{p}} \boldsymbol{V}(\boldsymbol{\theta_t}, t, \mathcal{G}; \boldsymbol{w}) \, \mathrm{d}\mathbf{B}_t$, where $\mathbf{B}_t$ is the standard Brownian motion, we can approximate it using the forward Euler scheme $\boldsymbol{\theta_{n+1}} = \boldsymbol{\theta_n} + \boldsymbol{V}(\boldsymbol{\theta_n}, n, \mathcal{G}; \boldsymbol{w})\Delta t + \sqrt{\frac{1-p}{p}} \boldsymbol{V}(\boldsymbol{\theta_n}, n, \mathcal{G}; \boldsymbol{w})\Delta \boldsymbol{W_n}$. The constructed Eq.16 is an order-0.5 weak approximation to it (Peter E. Kloeden, 1992) since it satisfies the following property:

$$
|\mathbb{E}(\Delta \hat{W}_n)| + |\mathbb{E}((\Delta \hat{W}_n)^2) - \Delta t| \leq K \Delta t^{1.5}
\tag{17}
$$

where $\Delta \hat{W}_n$ is the approximation to the original Gaussian random variable $\Delta \boldsymbol{W_n}$, which is $\boldsymbol{z_n}$ here. A weak order 0.5 approximation converges weakly to the Itô process in a rate proportional to $\Delta t^{1.5}$. Hence, distributionally, we can focus on Itô process instead regarding stability analysis.

## B  PROOF DETAILS

***Proof of Theorem 1.*** First, rewrite the parameter evolution equation (Eq. 4) as a shorthand expression:

$$
\mathrm{d}\boldsymbol{\theta_t} = \mu_t \, \mathrm{d}t + \sigma_t \, \mathrm{d}\mathbf{B}_t \,,
\tag{18}
$$

where $\mu_t, \sigma_t$ are square-integrable. The sub-network $U$ is a twice continuously differentiable function w.r.t. $\boldsymbol{\theta_t}$, denoted as $Y_t(\omega) = U_{\boldsymbol{\theta_t}}$. By Itô formula:

$$
\begin{aligned}
\mathrm{d}Y_t &= \sum_{i=1}^{d} \frac{\partial U_{\boldsymbol{\theta_t}}}{\partial \theta_i} \mathrm{d}\boldsymbol{\theta}_{i,t} + \frac{1}{2} \sum_{i,j=1}^{d} \frac{\partial^2 U_{\boldsymbol{\theta_t}}}{\partial \theta_i \theta_j} \mathrm{d}\boldsymbol{\theta}_{i,t} \mathrm{d}\boldsymbol{\theta}_{j,t} \\
&= \left( \sum_{i=1}^{d} \frac{\partial U_{\boldsymbol{\theta_t}}}{\partial \theta_i} \mu_{i,t} + \frac{1}{2} \sum_{i=1}^{d} \frac{\partial^2 U_{\boldsymbol{\theta_t}}}{\partial \theta_i^2} \sigma_{i,t}^2 \right) \mathrm{d}t + \left( \sum_{i=1}^{d} \frac{\partial U_{\boldsymbol{\theta_t}}}{\partial \theta_i} \sigma_{i,t} \right) \mathrm{d}\mathbf{B}_t \\
&:= \mu'_t \mathrm{d}t + \sigma'_t \mathrm{d}\mathbf{B}_t \,.
\end{aligned}
\tag{19}
$$

Given a Brownian path $\omega$ in advance, Denote the solution at time $t$ with initial condition $Y_s$ as $T_t(Y_s)(\omega)$. The Itô integral of the SDE gives the following stochastic integral equations for any $t1, t2 \geq 0$ and any initial $Y_0$ almost surely:

$$
\begin{aligned}
T_{t_1+t_2}(Y_0)(\omega) &= \int_0^{t_1+t_2} \mu_r' \, \mathrm{d}r + \int_0^{t_1+t_2} \sigma_r' \, \mathrm{d}\boldsymbol{B}_r(\omega) + Y_0 \\
T_{t_1}(Y_0)(\omega) &= \int_0^{t_1} \mu_r' \, \mathrm{d}r + \int_0^{t_1} \sigma_r' \, \mathrm{d}\boldsymbol{B}_r(\omega) + Y_0 \\
T_{t_2} T_{t_1}(Y_0)(\omega) &= T_{t_2}(T_{t_1}(Y_0)(\omega))(\omega) \\
&= \int_{t_1}^{t_1+t_2} \mu_r' \, \mathrm{d}r + \int_{t_1}^{t_1+t_2} \sigma_r' \, \mathrm{d}\boldsymbol{B}_r(\omega) + T_{t_1}(Y_0)(\omega)
\end{aligned}
\tag{20}
$$

It follows from the additive property of Itô integral that the solution operator $T$ is a strong semigroup:

$$
T_{t_1+t_2}(\omega)(Y_0) = T_{t_2} T_{t_1}(\omega)(Y_0) \text{ a.s.}, \quad T_0(\omega)(Y_0) = Y_0 \text{ a.s.}
\tag{21}
$$

A strong equality implies a weak equality (Øksendal & Øksendal, 2003), therefore the solution also satisfies weak semigroup property. □

***Proof of Theorem 3.*** Firstly, the task could be separated into two subtasks. We need to evaluate the perturbation's influence on $\boldsymbol{\theta_t}$ and $r_t$ separately.

$$
\limsup_{t \to \infty} \frac{1}{t} \log \| \begin{bmatrix} \boldsymbol{\varepsilon_t^\theta} \\ \varepsilon_t^r \end{bmatrix} \| \leq \limsup_{t \to \infty} \frac{1}{t} \log \| \begin{bmatrix} \boldsymbol{\varepsilon_t^\theta} \\ 0 \end{bmatrix} \| + \limsup_{t \to \infty} \frac{1}{t} \log \| \begin{bmatrix} \boldsymbol{0} \\ \varepsilon_t^r \end{bmatrix} \|
\tag{22}
$$

Theorem 2 has already proved the stability of Eq.10:

$$
\limsup_{t \to \infty} \frac{1}{t} \log \|\boldsymbol{\varepsilon_t^\theta}\| < 0
\tag{23}
$$

And hence $\exists \alpha > 0, T > 0$, s.t. $\forall t > T, \|\boldsymbol{\varepsilon_t^\theta}\| \leq e^{-\alpha t}$.

As is proved in Liu et al. (2019), it is said that $R_\Delta$ aligns with the assumptions if $R$ does, which means that $\|R_\Delta(\boldsymbol{\varepsilon_t^\theta}, t)\| \leq L(\|\boldsymbol{\varepsilon_t^\theta}\| + 1)$. $R_\Delta$ here is a scalar function, and the norm represents its absolute value. Substituting the above equations into the SDE we will get:

$$
\frac{\mathrm{d}\varepsilon_t^r}{\mathrm{d}t} = R_\Delta(\boldsymbol{\varepsilon_t^\theta}, t) \leq L(e^{-\alpha t} + 1)
\tag{24}
$$

By integrating this simple ODE, we obtain an upper bound on the perturbation's influence on $r_T$:

$$
\varepsilon_t^r \leq \varepsilon_0^r + \int_0^t L(e^{-\alpha t} + 1) \, \mathrm{d}t = \varepsilon_0^r + Lt + \frac{L}{\alpha}(1 - e^{-\alpha t})
\tag{25}
$$

Substituting Eq.25 into the objective function, we can conclude that $r_t$ is almost surely exponentially stable and thus prove that the solution to this SDE system is almost surely exponentially stable.

$$
\begin{aligned}
\limsup_{t \to \infty} \frac{1}{t} \log \|\varepsilon_t^r\| &\leq \limsup_{t \to \infty} \frac{1}{t} \log \|\varepsilon_0^r + Lt + \frac{L}{\alpha}(1 - e^{-\alpha t})\| \\
&\leq \limsup_{t \to \infty} \frac{1}{t} \log |Lt| \leq 0
\end{aligned}
\tag{26}
$$

□

**Proof of Theorem 4.** Firstly, since given continuous activation function, neural networks can generally be seen as continuous functions of network parameters, we can conclude that $\forall \varepsilon, \exists \varepsilon_\Theta$, if $\|\boldsymbol{\theta}_t^e - \boldsymbol{\theta}_t\| < \varepsilon_\Theta$, then $\|U_{\boldsymbol{\theta}_t^e} - U_{\boldsymbol{\theta}_t}\| < \varepsilon$.

Secondly, Theorem 3 proves a stricter version of stability of the evolution of $\boldsymbol{\Theta}_t$. Hence, given $\varepsilon_\Theta$, $\exists \delta_\Theta$, if $\|\boldsymbol{\Theta}_0^e - \boldsymbol{\Theta}_0\| < \delta_\Theta$, then $\|\boldsymbol{\Theta}_t^e - \boldsymbol{\Theta}_t\| < \varepsilon_\Theta$, and hence $\|\boldsymbol{\theta}_t^e - \boldsymbol{\theta}_t\| < \varepsilon_\Theta$. Conclusively, $\forall \varepsilon, \exists \delta_\Theta$, if $\|\boldsymbol{\Theta}_0 - \boldsymbol{\Theta}_0^e\| < \delta_\Theta, \|U_{\boldsymbol{\theta}_t^e} - U_{\boldsymbol{\theta}_t}\| < \varepsilon$. SINGER is stable during training.

$\square$

**Proof of Theorem 5.** Firstly, we attempt to approximate the Gradient Descent (GD) with Bernoulli noise as an SDE. GD could be represented as $\boldsymbol{\theta}_{n+1} = \boldsymbol{\theta}_n - \eta \nabla \mathcal{L}(\boldsymbol{\theta}_n)$, where $\eta$ is the learning rate. If we drop random elements of the gradient, we can get the following formula:

$$
\begin{aligned}
\boldsymbol{\theta}_{0,n+1} &= \boldsymbol{\theta}_{0,n} - \eta \nabla \mathcal{L}(\boldsymbol{\theta}_{0,n}) \odot \frac{\boldsymbol{\gamma}_{0,n}}{p} \\
&= \boldsymbol{\theta}_{0,n} - \eta \nabla \mathcal{L}(\boldsymbol{\theta}_{0,n}) - \eta \nabla \mathcal{L}(\boldsymbol{\theta}_{0,n}) \odot \left( \frac{\boldsymbol{\gamma}_{0,n}}{p} - \mathbf{I} \right)
\end{aligned}
\tag{27}
$$

If we make this representation continuous as in Appendix A, we can get the following SDE:

$$
\mathrm{d}\boldsymbol{\theta}_{0,\tau} = -\eta \nabla \mathcal{L}(\boldsymbol{\theta}_{0,\tau}) \, \mathrm{d}\tau - \eta \sqrt{\frac{1-p}{p}} \nabla \mathcal{L}(\boldsymbol{\theta}_{0,\tau}) \, \mathrm{d}\boldsymbol{B}_\tau
\tag{28}
$$

Given the conditions provided, Theorem 3 indicates that Eq.28 is stable. Following the proof in Theorem 4, $\forall \delta_\Theta, \exists \delta$, if $\|U_{\boldsymbol{\theta}_0} - U_{\boldsymbol{\theta}_0^e}\| < \delta$, then $\|\boldsymbol{\theta}_0^e - \boldsymbol{\theta}_0\| < \delta_\Theta$. Since we always set $r_0 = 0$, $\|\boldsymbol{\Theta}_0^e - \boldsymbol{\Theta}_0\| < \delta_\Theta$. In conclusion, SINGER is stable during testing. $\square$

## C   EQUATIONS AND EXPERIMENT SETTING DETAILS

### C.1   BACKWARD STOCHASTIC DIFFERENTIAL EQUATIONS (BSDE) METHOD

The BSDE (Han et al., 2018) method reformulates certain partial differential equations (PDEs) as backward stochastic differential equations (BSDEs), which can then be solved using numerical methods. The general form of a BSDE is:

$$
Y_t = g(X_T) + \int_t^T f(s, X_s, Y_s, Z_s) \, ds - \int_t^T Z_s^\top \, dW_s,
$$

where $X_t$ represents the stochastic process, $Y_t$ and $Z_t$ are adapted processes, and $g(X_T)$ is the terminal condition.

In our experiments, for each type of equation, we generate data by adjusting the terminal condition function $g(x)$ with different parameters. The BSDE solver is then used to solve these modified equations, providing us with a dataset of triplets $(t, x, u_{\mathrm{pred}})$, where $t$ represents time, $x$ is the state variable of the PDE, and $u_{\mathrm{pred}}$ is the predicted solution at time $t$.

### C.2   HAMILTON-JACOBI-BELLMAN LINEAR-QUADRATIC (HJB-LQ) EQUATION

#### C.2.1   EQUATION OVERVIEW

The Hamilton-Jacobi-Bellman (HJB) equation arises in control theory and describes the value function for an optimal control problem. In the classical linear-quadratic-Gaussian (LQG) control problem, the equation is given by:

$$
\frac{\partial u}{\partial t}(t, x) = \frac{1}{2}\sigma^2 \Delta u(t, x) - \lambda \|\nabla u(t, x)\|^2, \quad t \in [0, T], \ x \in \mathbb{R}^d,
$$

with the terminal condition $u(T, x) = g(x)$.

In our experiments, the terminal cost function is defined as:

$$g(x) = \ln\left(\frac{1 + k \cdot \|x\|^2}{2}\right),$$

where $k$ is the adjustable coefficient.

### C.2.2 PARAMETER SETTINGS

In our experiments, the following parameters were used for solving the HJB-LQ equation and for training the neural network to approximate the solution:

- **Total time of the equation:** total_time = 0.4
- **Number of time intervals for solving the PDE:** num_time_interval = 20
- **Range of the adjustable coefficient in the terminal condition** $g$: k_range = [0.5, 1.5]
- **Parameters for fitting the generated data using the neural network:**
    - Number of iterations for fitting: fit_n_iter = 20,000
    - Batch size during fitting: fit_batch_size = 64
    - Threshold for stopping fitting based on loss: fit_threshold = 1e-3
- **Parameters for training the neural network using our method:**
    - Dropout rate during training: train_dropout = 0.3
    - Batch size during training: train_batch_size = 64
    - Learning rate during training: train_learning_rate = 1e-4

### C.3 REACTION-DIFFUSION EQUATION

### C.3.1 EQUATION OVERVIEW

The reaction-diffusion equation models the behavior of a system undergoing both diffusion and reaction processes. The equation is given by:

$$\frac{\partial u}{\partial t}(t, x) = \frac{1}{2}\Delta u(t, x) + f(t, x, u, \nabla u), \quad t \in [0, T], \ x \in \mathbb{R}^d,$$

where the function $f$ is defined as:

$$f(t, x, u, \nabla u) = \min\left(1, \left(u - \kappa - 1 - \sin\left(\lambda \sum_{i=1}^{d} x_i\right) e^{\frac{\lambda^2 d(t-T)}{2}}\right)^2\right),$$

where $\lambda$ is a constant representing the wave frequency, $\kappa$ is a constant related to the initial conditions, and $T$ is the total time.

The terminal condition $g(x)$ for this reaction-diffusion equation is defined as:

$$g(x) = 1 + \kappa + \sin\left(\lambda \sum_{i=1}^{d} x_i\right) \cdot k,$$

where $k$ is the the adjustable coefficient.

### C.3.2 PARAMETER SETTINGS

In our experiments, the following parameters were used for solving the reaction-diffusion equation and training the neural network:

- **Total time of the equation:** total_time = 0.6
- **Number of time intervals for solving the PDE:** num_time_interval = 30

- **Range of the adjustable coefficient in the terminal condition** $g$**:** k_range = [0.5, 1.5]
- **Parameters for fitting the generated data using the neural network:**
  - Number of iterations for fitting: fit_n_iter = 30,000
  - Batch size during fitting: fit_batch_size = 64
  - Threshold for stopping fitting based on loss: fit_threshold = 1e-3
- **Parameters for training the neural network using our method:**
  - Dropout rate during training: train_dropout = 0.4
  - Batch size during training: train_batch_size = 64
  - Learning rate during training: train_learning_rate = 2e-4

### C.4 BURGERS-TYPE EQUATION

#### C.4.1 EQUATION OVERVIEW

The Burgers-type equation is a fundamental partial differential equation that models various physical phenomena such as fluid dynamics, gas dynamics, and traffic flow. The general form of the equation is given by:

$$\frac{\partial u}{\partial t}(t, x) = \frac{1}{2}\sigma^2 \Delta u(t, x) + \left(u - \frac{2 + d}{2d}\right)\sum_{i=1}^{d}\frac{\partial u}{\partial x_i}, \quad t \in [0, T], \ x \in \mathbb{R}^d$$

The terminal condition $g(x)$ for the Burgers-type equation is defined as:

$$g(x) = 1 - \frac{1}{1 + \exp\left(k \cdot t + \frac{1}{d}\sum_{i=1}^{d} x_i\right)},$$

where $k$ is the the adjustable coefficient.

#### C.4.2 PARAMETER SETTINGS

In our experiments, the following parameters were used for solving the Burgers-type equation and training the neural network to approximate the solution:

- **Total time of the equation:** total_time = 0.3
- **Number of time intervals for solving the PDE:** num_time_interval = 30
- **Range of the adjustable coefficient in the terminal condition** $g$**:** k_range = [0.8, 1.3]
- **Parameters for fitting the generated data using the neural network:**
  - Number of iterations for fitting: fit_n_iter = 30,000
  - Batch size during fitting: fit_batch_size = 64
  - Threshold for stopping fitting based on loss: fit_threshold = 1e-3
- **Parameters for training the neural network using our method:**
  - Dropout rate during training: train_dropout = 0.3
  - Batch size during training: train_batch_size = 64
  - Learning rate during training: train_learning_rate = 1e-4

### C.5 PRICING DEFAULT RISK EQUATION

#### C.5.1 EQUATION OVERVIEW

The Pricing Default Risk equation is a variant of the Black-Scholes equation that models the pricing of financial derivatives under default risk. The equation is given by:

$$\frac{\partial u}{\partial t}(t, x) = \frac{1}{2}\sigma^2 x^2 \Delta u(t, x) + (-(1 - \delta) \cdot \text{piecewise\_linear}(u) - r) \cdot u, \quad t \in [0, T], \ x \in \mathbb{R}^d$$

where the piecewise linear term is expressed as:
$$\text{piecewise\_linear}(u) = \text{ReLU}(\text{ReLU}(u - v_h) \cdot \text{slope} + \gamma_h - \gamma_l) + \gamma_l,$$
where $\delta$ represents the recovery rate after default, $r$ is the risk-free interest rate, $v_h$ is a threshold parameter, $\gamma_h$ and $\gamma_l$ are constants representing the different regimes of the default intensity, and slope determines the linear transition between regimes.

The terminal condition $g(x)$ for the Pricing Default Risk equation is defined as:
$$g(x) = \min(x) \cdot k,$$
where $k$ is the the adjustable coefficient.

### C.5.2 PARAMETER SETTINGS

In our experiments, the following parameters were used for solving the Pricing Default Risk equation and training the neural network to approximate the solution:

- **Total time of the equation:** total_time = 0.2
- **Number of time intervals for solving the PDE:** num_time_interval = 20
- **Range of the adjustable coefficient in the terminal condition** $g$**:** k_range = [0.5, 1.0]
- **Parameters for fitting the generated data using the neural network:**
    - Number of iterations for fitting: fit_n_iter = 30,000
    - Batch size during fitting: fit_batch_size = 64
    - Threshold for stopping fitting based on loss: fit_threshold = 1e-3
- **Parameters for training the neural network using our method:**
    - Dropout rate during training: train_dropout = 0.4
    - Batch size during training: train_batch_size = 64
    - Learning rate during training: train_learning_rate = 1e-4

### C.6 SINE-GORDON EQUATION

### C.6.1 EQUATION OVERVIEW

The Sine-Gordon equation is a second-order nonlinear partial differential equation frequently utilized in the field theory, string theory, and solitons. The equation is given by:

$$\frac{\partial u}{\partial t}(t, x) = \Delta u(t, x) + \sin(u(t, x)) \quad t \in [0, T], \ x \in \mathbb{R}^d$$

The terminal condition $g(x)$ for Sine-Gordon equation is defined as:

$$g(x) = \frac{5}{10 + 2\|x\|^2}.$$

### C.6.2 PARAMETER SETTINGS

In our experiments, the following parameters were used for solving the Sine-Gordon equation and training the neural network to approximate the solution:

- **Total time of the equation:** total_time = 0.3
- **Number of time intervals for solving the PDE:** num_time_interval = 20
- **Parameters for fitting the generated data using the neural network:**
    - Number of iterations for fitting: fit_n_iter = 5,000
    - Batch size during fitting: fit_batch_size = 100
    - Threshold for stopping fitting based on loss: fit_threshold = 1e-3
- **Parameters for training the neural network using our method:**
    - Dropout rate during training: train_dropout = 0.1
    - Batch size during training: train_batch_size = 100
    - Learning rate during training: train_learning_rate = 1e-3

### C.7 ALLEN-CAHN EQUATION

#### C.7.1 EQUATION OVERVIEW

The Allen-Cahn equation is reaction-diffusion equation that models various physical phenomena such as phase separation, interfacial dynamics and pattern formation. The equation is given by:

$$\frac{\partial u}{\partial t}(t, x) = \Delta u(t, x) + u(t, x) - [u(t, x)]^3 \quad t \in [0, T], \, x \in \mathbb{R}^d$$

The terminal condition $g(x)$ for Allen-Cahn equation is defined as:

$$g(x) = \frac{5}{10 + 2\|x\|^2}.$$

#### C.7.2 PARAMETER SETTINGS

In our experiments, the following parameters were used for solving the Allen-Cahn equation and training the neural network to approximate the solution:

- **Total time of the equation:** total_time = 0.3
- **Number of time intervals for solving the PDE:** num_time_interval = 20
- **Parameters for fitting the generated data using the neural network:**
    - Number of iterations for fitting: fit_n_iter = 5,000
    - Batch size during fitting: fit_batch_size = 100
    - Threshold for stopping fitting based on loss: fit_threshold = 1e-3
- **Parameters for training the neural network using our method:**
    - Dropout rate during training: train_dropout = 0.1
    - Batch size during training: train_batch_size = 100
    - Learning rate during training: train_learning_rate = 1e-3

## D RELATED WORK ON NEURAL PDE SOLVERS IN LOW-DIMENSIONS

The neural-network-based approaches for low dimensional PDEs can be mainly divided into two categories: physics-informed network and data-driven operator(Karniadakis et al., 2021). Specifically, Physics-Informed Neural Networks (PINNs) (Raissi et al., 2019) leverage neural networks to approximate the solution of a PDE and enforce boundary loss and PDE residual loss at a few selected points. This approach has various extensions such as weak form De Ryck et al. (2024), adaptive sample weight McClenny & Braga-Neto (2020) and integral loss (Saleh et al., 2023). These methods may suffer from slow convergence and re-training requirements for new initial conditions. Alternatively, DeepONet (Chen & Chen, 1995; Lu et al., 2021) leverages the universal approximation theorem of infinite-dimensional operators and directly fits the PDE solution operators in a data-driven manner, solving a family of PDEs with a single model. However, this approach requires a large amount of data for training and may not generalize well to new initial conditions. Physics-informed neural operators (PINO) Li et al. (2024) is a hybrid approach incorporating data and PDE residuals in the loss function. There exist many neural operators with other architectures, such as Fourier transform layer (Li et al., 2020), and graph neural networks (GNNs)(Alet et al., 2019). The existing GNN-based models describe irregular spacial grids and their relative position as graph (Lötzsch et al., 2022; Horie & Mitsume, 2022; Bryutkin et al., 2024).

## E FUTURE WORK AND BROAD IMPACT

Future work will explore extending this method to even higher dimensions and more complex systems, potentially incorporating additional techniques of data sampling and training. A potential limitation of our approach is the generalization among PDE parameters, which could be addressed

by incorporating parameters into the model inputs and modifying the training loss. Another limitation is that current model is confined to continuous solution, which can be extended to discontinuous cases by adopting weak form residual.

**Broader Impact**: The proposed method has the potential to impact a wide range of scientific applications, including quantum mechanics, quantum chemistry and financial mathematics. The ability to learn high-dimensional PDEs solution operator, instead of solving a single problem instance, could lead to significant advancements in these fields. In addition, the generalization ability of the proposed model might pave the way for pre-trained foundation models of high-dimensional PDEs.

