# OpenReview forum: "SINGER: Stochastic Network Graph Evolving Operator for High Dimensional PDEs"
_ICLR.cc/2025/Conference — ICLR 2025 Poster_

### Official Review · Reviewer_kUCH · 2024-10-31

**Soundness:** 3
**Presentation:** 4
**Contribution:** 2
**Rating:** 5
**Confidence:** 3

**Summary:**

The authors introduce StochastIc Network Graph Evolving operatoR (SINGER), a novel framework for learning the evolution operator of high-dimensional partial differential equations (PDEs). SINGER employs a sub-network to approximate the initial solution and stochastically evolves its parameters over time using a graph neural network to approximate later solutions. Designed to inherit key properties such as graph topology, semigroup properties, and stability, SINGER comes with theoretical guarantees of performance. Numerical experiments on eight PDEs across various dimensions show that SINGER outperforms existing methods in most cases and generalizes effectively to new conditions and parameters.

**Strengths:**

1. The presentation is clear and concise, and the problem is clearly illustrated. Although if I am correct, the network solves the PDEs at fixed collocation points, it would be great to point that out somewhere.
2. The empirical results are clear.
3. The theory of the method are sound and extensive.

**Weaknesses:**

1. While the empirical results cover most important PDE solvers, it does not compare with the SOTA methods, and merely compare to the vanilla ones. What's more, methods such as neural ordinary differential equations (NODEs) are not meant for solving partial differential equations, it seems a bit unfair to compare to them.
2. For instance, the proposed method resides in the realm of hypernetworks, it would be nice to see a hypernetwork of PINN or DeepONet.

**Questions:**

1. Why is it beneficial to have the stochasticity? Usually they hurts the surrogate performances, is there an ablation on the noise level to illustrate this issue?
2. The test cases are all very smooth PDEs, any reason why this is the case? Is it possible to either prove the ability on non-smooth PDEs or show with empirical results?

---

> ### Author Response · Authors · 2024-11-23
> **Rebuttal (Part 1/2)**
>
> We sincerely thank the reviewer for the insightful and encouraging feedback. We are pleased that the reviewer recognizes the novelty of our proposed SINGER framework for learning the evolution operator of high-dimensional partial differential equations (PDEs). Below, we provide detailed responses and clarifications to address the reviewer’s comments and questions.
>
> ---
> **W1: Baseline is not SOTA. It would be nice to see a hypernetwork of PINN or DeepONet.**
>
> **A1.** The problem of baselines can be discussed in three folds:
> - **SOTA Baselines**
>     1. NODE (Gaby & Ye, 2024, see reference [1]) is the SOTA for neural operators targeting high-dimensional PDEs, to the best of our knowledge. Currently, no studies have directly applied low-dimensional PDE neural operators (e.g., DeepONet or FNO) to high-dimensional problems. This is primarily due to their reliance on substantial reference data for training, with the required data volume increasing exponentially with spatial dimensions (the so-called curse of dimensionality). The training data is typically generated using numerical solvers, but high-dimensional PDEs lack accurate and efficient numerical solvers, making data generation challenging. Even if sufficient training data could be generated, the training time and memory requirements would be prohibitively large.
>     2. PINN is not a valid baseline of neural operators.  Well-recognized neural operator papers like DeepONet[2] and FNO[3] do not compare PINN as a baseline.  The main reason might be that PINNs are not operators by definition. Since a PINN trained on one initial condition does not generalize to other conditions, i.e. it needs re-training for each new input. In addition, the PINN solving is much more time-consuming than a neural operator, making the comparison impractical and unfair. for example, the DeepBSDE solver [4] (a variant of PINN) solves a high-dimension PDE in 1 minute, while our trained SINGER solves it in about 0.1 seconds.
>
>
> - **Hyper-PINN-Nerual-Operator (PINO)**
> To close the gap, in our paper, we designed a tailored Hyper-PINN-Nerual-Operator (PINO) as an additional baseline for the high-dimensional PDEs. Compared to standard neural operators, PINO introduces two major advancements. First, it replaces data loss with physics-informed loss, removing the dependence on training data. Second, it employs a hyper-network to generate parameters for a sub-network, which then produces a parameterized solution. This approach avoids directly outputting solutions at spatial sampling points, thus mitigating the exponential growth of network size with increasing spatial dimensions. These extensions make network training feasible.
>
>
> - **PINO Performance**
> Although PINO is a feasible model, it falls short of SOTA theoretic support and empirical performance. From a model design perspective, PINO does not adhere to the semigroup assumption, whereas our proposed SINGER does (see also the direct evidence from Table 5). In terms of performance, experimental results on the Heat and HJB problems (Table 2) indicate that PINO consistently underperforms compared to NODE and our SINGER across all tested dimensions. In some cases, it even produces entirely incorrect results. For example, for the HJB problem with d=15 and d=20, the relative error of PINO approaches or exceeds 1, comparable to the performance of a dummy model producing zero outputs.  These results suggest our adapted PINO still lags behind the available peer methods.
>
>
> ---
> **Q2: What are the benefits of stochasticity? Need ablation on the noise level.**
>
> **A2.** Moderate noise, compared to no noise, can enhance both stability and performance. Theoretical support for this claim is provided by Theorems 2–4, while experimental evidence is presented in Table 2 and Figure 4. Adding noise accelerates convergence and improves the final results, whereas the absence of noise often leads to training collapse.
>
> However, the magnitude of noise does not improve performance indefinitely; excessive noise can be detrimental. In our paper, we fixed the noise level at 0.1. To further investigate, we conducted additional experiments testing the relative error of the heat equation under different noise levels (see Table A2 below). The results indicate that as the noise level increases from zero, the error initially decreases and then begins to rise. This behavior likely reflects a trade-off between the stability introduced by noise and the information loss it causes.
>
>
> > Table A2. SINGER Relative Error on Heat Equation under various noise levels.
>
> | Noise Level | 0      | 0.05   | 0.1    | 0.2    | 0.4    |
> | ----------- | ------ | ------ | ------ | ------ | ------ |
> | Dim=5       | 0.0137 | 0.0106 | 0.0045 | 0.0063 | 0.0162 |
> | Dim=10      | 0.0294 | 0.0056 | 0.0041 | 0.0097 | 0.0246 |
>
> ---

---

> > ### Author Response · Authors · 2024-11-23
> > **Rebuttal (Part 2/2)**
> >
> > **Q3. Why are all test cases smooth PDEs? Please either prove the ability on non-smooth PDEs or show empirical results.**
> >
> > **A3.** Thanks for your valuable question. This question can be answered in two folds:
> >
> > - **Reason for smooth PDEs.**
> > Existing methods for solving high-dimensional PDEs focus mainly on continuous problems, and we also limit our tests to continuous high-dimensional PDEs. Here are possible reasons:
> >     1. The test data is generated using existing numerical methods, which generally cannot handle high-dimensional discontinuous PDEs, making it infeasible to create a test dataset or metric.
> >     2. Discontinuous PDEs in scientific problems can sometimes be transformed into higher-dimensional continuous problems by increasing the dimensionality. For example, discontinuous non-linear hyperbolic equations (e.g., Burgers’ equation) can be reformulated as continuous high-dimensional PDEs using the level set method (see reference [5]).
> >     3. Solutions to discontinuous PDEs are typically defined only in the sense of weak solutions, which require introducing an additional high-dimensional integral, significantly increasing computational costs. Addressing this challenge remains a promising direction for future research.
> >
> > - **Empirical Results.**
> > We provide a numerical experiment to test SINGER's ability to approximate discontinuous solutions. For example, when the viscosity coefficient $\sigma^2 = 0$ in Burgers’ equation, the solution exhibits discontinuities (shock waves). By letting $\sigma^2 \to 0$, we can approximate the discontinuous solution. This approach, known as the vanishing viscosity method, is widely used in the analysis and numerical computation of discontinuous PDEs. We tested SINGER's relative error for the 5-dimensional Burgers’ equation under varying values of $\sigma$, with the results summarized in Table A3 below. These results demonstrate that SINGER is capable of learning vanishing viscosity solutions with high accuracy.
> >
> > > Table A3. SINGER for viscosity solution of Burgers Equation.
> >
> > | $\sigma^2$     | 1      | 1e-2   | 1e-4   | 1e-6   |
> > | -------------- | ------ | ------ | ------ | ------ |
> > | Relative Error | 0.0021 | 0.0021 | 0.0024 | 0.0025 |
> >
> > ---
> > **S1: Does the network solve the PDEs at fixed collocation points?**
> >
> > **A4.** No, the collocation points are randomly sampled from the entire domain of $x$ without repeat.
> >
> > ---
> >
> > **Claim of Contributions.**
> >
> > In this paper, we identify and formalize three key assumptions of an ideal PDE solution operator: graph topology, semigroup property, and stability. Then we propose SINGER consists of graph network, continuous evolution, and stochastic noise, corresponding to the three assumptions, with theoretical and empirical justification. Finally, SINGER outperforms competitive baselines on 8 PDEs and 5-20 dimensions.
> >
> > In the current review, we are glad that reviewers appreciate our work from multiple perspectives. Reviewer kUCH marks our **presentation as excellent**. Reviewer 8dkd recognizes the **quality of our motivation** and **soundness of our theoretical analysis and computation result**. Reviewer mPx4 remarked that our **central idea is novel**.
> >
> > ---
> > References:
> >
> > [1] Gaby, Nathan, and Xiaojing Ye. "Approximation of Solution Operators for High-dimensional PDEs." arXiv preprint arXiv:2401.10385 (2024).
> >
> > [2] Lu, Lu, et al. "Learning nonlinear operators via DeepONet based on the universal approximation theorem of operators." Nature machine intelligence 3.3 (2021): 218-229.
> >
> > [3] Li, Zongyi, et al. "Fourier neural operator for parametric partial differential equations." arXiv preprint arXiv:2010.08895 (2020).
> >
> > [4] Han, Jiequn, Arnulf Jentzen, and Weinan E. "Solving high-dimensional partial differential equations using deep learning." Proceedings of the National Academy of Sciences 115.34 (2018): 8505-8510.
> >
> > [5] Cheng, Li-Tien, Hailiang Liu, and Stanley Osher. "Computational high-frequency wave propogation using the level-set method with applications to the semi-classical limit of the Schrödinger equations." (2003): 593-621.
> >
> >
> > ---

---

> > > ### Comment · Reviewer_kUCH · 2024-11-24
> > >
> > > Thank you for your detailed response, I will raise my score to 5.

---

> > > > ### Author Response · Authors · 2024-11-26
> > > > **Thank You for the Updated Review**
> > > >
> > > > Dear Reviewer kUCH,
> > > >
> > > > We hope this message finds you well. We sincerely thank you for taking the time to review my work and for your thoughtful comments. We are especially grateful for your decision to increase the score, and we truly appreciate the constructive feedback that led to this improvement.
> > > >
> > > > Thank you once again for your time and effort. Please let us know if there are any new issues or further suggestions.
> > > >
> > > > Best regards,
> > > >
> > > > SINGER Authors

---

### Official Review · Reviewer_mPx4 · 2024-10-31

**Soundness:** 3
**Presentation:** 3
**Contribution:** 3
**Rating:** 6
**Confidence:** 3

**Summary:**

The paper proposes to solve time-dependent PDEs with a neural network model $U$that approximates the PDE solutions, where the neural network parameters are solutions to a system of stochastic differential equations with a drift term modeled by a Graph Neural Network on a graph induced by the architecture of $U$.

While I am unfamiliar with the literature, I think the idea is interesting, especially with the mathematical reasons discussed in the paper. Upon careful inspection, it seems that this work extends NODE with stochastic noise and GNN architecture for the control vector $V$, where they used tools from Liu et al. 2019 (which is a paper that was rejected in ICLR 2020). My slight reservation with this work is due to my confusion as to why the paper Liu et al. 2019 paper has not been published and the work NODE also seems to be under review. However, these facts should not be the basis for rejecting this work. Based on my reading, the math part of this work is rather trivial and I am not against the paper for publication after the authors satisfactorily answer my following questions.

**Strengths:**

The central idea of using GNN for modeling the control seems to be novel.

**Weaknesses:**

This writing needs more clarifications.

**Questions:**

1. Looking at Eq. (4), I assumed that the PDE that is to be solved is $u_t = F(U)$? Is there any restriction on $F$ (e.g., class of PDEs) for this approach to work well or to fail?
2. Can the authors clarify or give a detailed explanation on a sub-network of $U$? At this point, I am just reading the paper as if the entire architecture of $U$ forms a graph, and the parameters in the neural networks are denoted by $\theta_t$. How do you set $V,E$ a priori? If it is a fully connected network, does it mean $E$ is always 1 between any pair of vertices? How do you choose a sub-network of $U$? How critical is the performance under variations of sub-networks (what if you use the entire full network)?
3. How do you specify $N$ in (3)? I suspect you need to set it to $N^2>2L$ based on Theorem 1, where $L$ is the Lipschitz constant of $V$, which needs to be estimated in the optimization procedure?
4. I am not familiar with NODE or PINO, so it is hard for me to say that the comparison in numerical experiments is fair. How does this work compared to any other methods to solve such a high-dimensional PDEs that are cited in the references (such as DeepRitz, PINN).
5. As noted in p.6, the semigroup property depends on the random seed in the solution of SDEs. I think stating that the method satisfies the semigroup property in Table 1 is overselling. Upon inspecting the setup for the stability (Eq (6)) in the manuscript, the analysis is performed under the assumption that the same path of Brownian noise is used. I think one can avoid this assumption by stating a weaker result (convergence in distribution instead of almost surely convergence). If my conjecture is correct, there is more theoretical work to satisfy the semigroup property in a weaker sense instead of stating "approximately satisfied". Table 5 also suggests that the semigroup is weakened after training (unless I misunderstood the reported values in this table)?
6. Minor: Should $u_{\theta_t}$ below Eq. (4) be $U_{\theta_t}$?

---

> ### Author Response · Authors · 2024-11-23
> **Rebuttal (Part 1/2)**
>
> We sincerely thank the reviewer for the thoughtful feedback and for recognizing the novelty of our approach in solving high-dimensional time-dependent PDEs using a neural network model driven by stochastic differential equations and graph neural networks. Below, we carefully address the reviewer’s specific questions and concerns:
>
> ---
> **Q1. What is the PDE to be solved? Is there any restriction on the class of PDEs?**
>
> **A1.** We have updated the problem setting in Eq. 3 in the new draft. The class (e.g. elliptic, hyperbolic, and parabolic class) of PDEs is not restricted. Currently, we only consider time-dependent PDEs with smooth solutions. In the future, we will extend the algorithm to discontinuous solutions by applying weak form residual.
>
> ---
> **Q2. Clarify on the sub-network U and G(V,E).  If it is a fully connected network (FCN), does it mean E is always 1 between any pair of vertices? How do you set it a priori? How critical is the performance under variations of sub-networks (what if you use the entire full network)?**
>
> **A2.** The questions of G and U can be answered in three parts:
>
> - **Construct G from U.**
> The graph nodes V are parameters of U, and graph edges are connections of parameters. Two parameters are connected if they handle the input and output of the same intermediate variable (i.e. neuron). The parameters in the same layer is always not connected. For a toy example, consider a 1-width network $U(x)=c\sigma(ax + b) + d$, the resulting graph G is $a \to b \to c \to d$, which degenerates to a linked list. This example is a special case of FCN, and the resulting E is not always 1, i.e. G is not a full graph. In fact, one can observe that the G is always a bipartite graph.
>
> - **Configure U a priori.**
> For Heat and HJB equations, we follow the network configuration in [1], which is a 2-layer FCN with boundary value function encoded in activation function (Eq. 14 and 15). For the other 6 PDEs, we use 2-layer FCN with tanh activation. The configuration of FCN is for simplicity.
>
> - **Sensitivity.**
> To show the sensitivity of network configuration, we tested the Heat equation as shown in Table A2. One can observe that the -1 Layer (i.e. single layer FCN) is worse than the baseline, while +1 Layer and +10/-10 Width all outperform the baseline, indicating that our current network configuration can be further optimized. The variations above are all bipartite graphs. We also tested the full graph model you mentioned, but it only produces degenerated results, as shown in the last column in the table.
>
>
>
> > Table A2. Sub-network configure sensitivity
> >
> | Modification   | -      | -1 Layer | +1 Layer | -10 Width | +10 Width | Full Graph |
> | -------------- | ------ | -------- | -------- | --------- | --------- | ---------- |
> | Relative Error | 0.0045 | 0.0507   | 0.0033   | 0.0033    | 0.0032    |  1.0425  |
>
> ---
>
> **Q3. How to specify noise N? Is it based on the Lipschitz constant L of V?**
>
> **A3.** Setting $N > \sqrt{2L}$ as you mentioned is a theoretical method for setting noise level. However, in practice, it is challenging to estimate $L$ since it requires a high-dimensional search and changes during training. In our experiment, we set the noise level N=0.1 constant for simplicity. Below we provide a comparison of different settings of $N$. It is clear that the relative error first decreases and then increases as the noise level increases. This phenomenon might result from a trade-off of stability and information loss due to noise.
>
> > Table A3. SINGER Relative Error on Heat Equation under various noise levels.
>
> | Noise Level | 0      | 0.05   | 0.1    | 0.2    | 0.4    |
> | ----------- | ------ | ------ | ------ | ------ | ------ |
> | Dim=5       | 0.0137 | 0.0106 | 0.0045 | 0.0063 | 0.0162 |
> | Dim=10      | 0.0294 | 0.0056 | 0.0041 | 0.0097 | 0.0246 |
> |             |        |        |        |        |        |

---

> ### Author Response · Authors · 2024-11-23
> **Rebuttal (Part 2/2)**
>
> ---
>
> **Q4. Fairness of NODE and PINO baselins. How does SINGER compare to high-dim PDE solvers such as DeepRitz and PINN?**
>
> **A4.** The baselines can be discussed in two folds.
>
> - **PINNs.**
> PINN, including its high-dim variants such as DeepRitz and DeepBSDE, is not a valid baseline of neural operators.  Well-recognized neural operator papers like DeepONet[2] and FNO[3] do not compare PINN as a baseline.  The main reason might be that PINNs are not operators by definition. Since a PINN trained on one initial condition does not generalize to other conditions, i.e. it needs re-training for each new input. In addition, the PINN solving is much more time-consuming than a neural operator, making the comparison impractical and unfair. for example, the DeepBSDE solver [4] solves a 20-dimensional PDE in about 1 minute, while our trained SINGER solves it in about 0.1 seconds.
>
> - **Neural Operators.**
> NODE [1] is the SOTA for neural operators targeting high-dimensional PDEs, to the best of our knowledge. In addition, we designed a baseline of Hyper-PINN-Nerual-Operator (PINO), which can be regarded as a special kind of PINN. Although PINO is a feasible model, it falls short of SOTA theoretic support and empirical performance. From a model design perspective, PINO does not adhere to the semigroup assumption, whereas our proposed SINGER does. In terms of performance, experimental results on the Heat and HJB problems (Table 2) indicate that PINO consistently underperforms compared to NODE and our SINGER across all tested dimensions.
>
> ---
> **Q5. More theoretical work is needed for the semigroup property in a weaker sense (in distribution instead of almost surely). Does Table 5 suggest that the semigroup is weakened after training?**
>
> **A5.** Thank you for the valuable suggestion. In the updated draft, we have clarified the strong (pathwise or almost surely) and weak (identical distribution) stability in Definition 1, and prove that SINGER satisfies both strong and weak stability in Appendix B. The proof follows naturally from the properties of the Itô formula and Itô integral.
>
> Besides, Table 5 seemingly suggests that the semigroup is weakened in the numerical sense after training. However, the relative error only increases to 1e-10 (1 out of 10 billion), which is usually regarded as numerically insignificant.
>
>
> ---
> **Q6. Minor: typo of $u$ and $U$.**
>
> **A6.** Thank you for pointing out that, we have fixed all typos about $u$ and $U$.
>
> ----
> Reference:
>
> [1] Gaby, Nathan, and Xiaojing Ye. "Approximation of Solution Operators for High-dimensional PDEs." arXiv preprint arXiv:2401.10385 (2024).
>
> [2] Lu, Lu, et al. "Learning nonlinear operators via DeepONet based on the universal approximation theorem of operators." Nature machine intelligence 3.3 (2021): 218-229.
>
> [3] Li, Zongyi, et al. "Fourier neural operator for parametric partial differential equations." arXiv preprint arXiv:2010.08895 (2020).
>
> [4] Han, Jiequn, Arnulf Jentzen, and Weinan E. "Solving high-dimensional partial differential equations using deep learning." Proceedings of the National Academy of Sciences 115.34 (2018): 8505-8510.

---

> > ### Author Response · Authors · 2024-11-25
> > **Gentle Follow-up on Review Status**
> >
> > Dear Reviewer mPx4,
> >
> > We hope you are doing well. We wanted to kindly follow up regarding your review of SINGER. We sincerely appreciate the time and effort you have dedicated to evaluating our work and providing valuable feedback.
> >
> > If possible, we would be grateful if you could update the score based on our responses to your comments. Your revised assessment will be incredibly helpful in moving the process forward. Of course, if there are any remaining questions or clarifications needed from us, we would be more than happy to assist promptly.
> >
> > Thank you once again for your support and contributions to this review process. We look forward to hearing from you soon.
> >
> > Best regards,
> >
> > SINGER Authors

---

> ### Author Response · Authors · 2024-11-26
> **Response Reminder**
>
> As we near the end of the discussion period, we would like to thank the reviewer again for the comments and provide a gentle reminder that we have posted a response to your comments. May we please check if our responses have addressed your concerns and improved your evaluation of our paper?

---

> > ### Comment · Reviewer_mPx4 · 2024-11-26
> >
> > You have addressed my concerns; I have no problem of raising the score to publishable.

---

> > > ### Author Response · Authors · 2024-11-26
> > >
> > > Dear reviewer,
> > >
> > > Thank you very much for your feedback. We are pleased to hear that you now consider the manuscript to be at a publishable level.
> > >
> > > However, We noticed that your final score is below the publishable threshold, which did not reflect the change in your assessment. We were wondering if it would be possible to reconsider the rating to better align with your positive evaluation of the revised manuscript.
> > >
> > > We truly appreciate your help and the opportunity to improve the work.
> > >
> > > Best regards,
> > >
> > > SINGER Authors

---

### Official Review · Reviewer_8dkd · 2024-11-09

**Soundness:** 3
**Presentation:** 3
**Contribution:** 3
**Rating:** 8
**Confidence:** 2

**Summary:**

In this work the authors propose a stochastic graph neural network based framework for solving high dimension partial differential equations (PDE).  Neural PDE solvers that perform well in low-dimensional settings do not generalize to high-dimensional settings and hence require special attention. However, existing methods in the high dimension regime suffer from instability and do not generalize to different types of PDEs. To overcome these drawback, the authors propose the SINGER framework.

The SINGER model uses a GNN to approximate the solution of the PDE at the initial time step and then stochastically evolves the network parameters over time according to a GNN-driven ODE. This network is then used to approximate the solution at later time steps. The structure enables permutability of neurons within a layer, this enhances the models capability to generalize. To combat the issue of instability in the evolution process, SINGER introduces noise during the training step.  Further more, SINGER is designed to satisfy three key assumptions: graph topology, semigroup property and stability. The authors theoretically and empirically verify that their proposed framework satisfies these assumptions. Finally, SINGER is validated on 8 benchmark PDEs and its performance is compared with state-of-the-art methods.

**Strengths:**

1) The authors identify gaps in current literature and do a good job in motivating their research.

2) The contributions of their framework are explained clearly.

3) Theoretical analysis of stability, graph topology and semigroup property of SINGER

4) Computational results are positive

**Weaknesses:**

Section 3: The authors could do a better job in explaining what the 3 assumptions signify and why They are important for solving PDEs.

**Questions:**

(page 1) The 3 contributions of the paper can be rewritten clearly and explained in more detail.

The authors can consider briefly explaining their theoretical contributions earlier in the paper.

(Table 3) Why not replace 'ours' with SINGER

---

> ### Author Response · Authors · 2024-11-23
>
> We sincerely thank the reviewer for the detailed and thoughtful feedback. We are pleased that the reviewer recognizes the importance of addressing the challenges in solving high-dimensional partial differential equations (PDEs) and appreciate our proposed SINGER framework for tackling these issues.  Below, we provide detailed responses and clarifications to address the reviewer's comments.
>
> ---
> **Q1. The meaning and importance of 3 assumptions**
>
> **A1.** These assumptions stem from the fundamental properties of PDEs and neural networks. Existing PDE neural solvers do not inherently satisfy these properties (see Table 1). In contrast, we explicitly encode these assumptions into the network, allowing them to serve as prior knowledge to help the network learn more accurately.
>
> - **Graph Topological Property**
>   Neural network solvers for high-dimensional PDEs typically parameterize the target function $u$ using a neural network $u_\theta$. The internal computation of $u_\theta$ forms a graph where the parameters $\theta$ represent the nodes. For general network architectures, the corresponding graph is invariant to permutation (re-indexing of nodes). Additionally, random removal of a node, such as through dropout, does not drastically alter the network output. Networks that do not follow this assumption tend to be sensitive to input noise (see the comparative experiments in Table 5). However, when solving PDEs, we desire that the solution be robust to noise, which motivates encoding this property into the model.
>
> - **Semigroup Property**
>   This property arises from the physical interpretation of PDEs, where the time evolution of the solution is independent. Similar to the Markov property, the future state depends only on the current state, with no memory of the past. Encoding this prior to the network can prevent it from learning "non-physical" solutions.
>
> - **Stability**
>   Stability is another crucial property that originates from the well-defined nature of PDEs. In general, one typically considers solving well-posed PDEs, i.e., those for which a solution exists, is unique, and continuously depends on the initial and boundary conditions. The stability we define refers to the continuous dependence of the PDE solution on the initial condition. Stability is of paramount importance in both theoretical analysis and numerical solutions of PDEs. For example, in our experiments, models without stability would collapse during training (see Table 2).
>
>
>
>
> ---
> **Q2. Writing Modification**
>
> **A2.** We rewrite the contribution (section 1), the motivation (section 3), and the table row name (table 3), according to your suggestions. The new version of the draft is uploaded, and the modified parts are highlighted in blue.

---

> > ### Author Response · Authors · 2024-11-25
> > **Follow-up on Review Feedback**
> >
> > Dear Reviewer 8dkd:
> >
> > We hope this message finds you well. We are writing to follow up on our previous responses. Do they provide enough information to better justify your confidence in the manuscript? If there are any remaining questions or areas needing clarification, please feel free to let us know—we are more than happy to address them promptly.
> >
> > Thank you once again for your valuable time and expertise. We look forward to hearing from you.
> >
> > Best regards,
> >
> > SINGER Authors

---

> > > ### Comment · Reviewer_8dkd · 2024-11-26
> > >
> > > Thank you for the clarification, my questions have been addressed.

---

### Meta-Review · Area_Chair_cJPf · 2024-12-17

**Metareview:**

The paper introduces SINGER (Stochastic Network Graph Evolving Operator), a novel framework for learning the evolution operator of high-dimensional partial differential equations (PDEs). SINGER uses a sub-network to approximate initial solutions and evolves its parameters stochastically over time using a graph neural network. The method is designed to maintain key properties of the parametric solution operator, such as graph topology, semigroup structure, and stability, with theoretical guarantees. Numerical experiments on 8 PDEs across various dimensions (5–20) demonstrate that SINGER outperforms most baselines generalizes effectively to unseen conditions, dimensions, sub-network widths, and time steps.

Most of the reviewers believe that the paper makes substantial contributions, and the AC also agrees with this assessment.

**Additional Comments On Reviewer Discussion:**

Three reviewers have evaluated the paper, and their overall assessment is positive. I agree with their evaluation and believe the paper offers a strong contribution with compelling results.

All reviewers raised a few technical questions, which the authors have addressed satisfactorily. I strongly recommend that the authors incorporate these remarks in the final version.

---

### Decision · Program_Chairs · 2025-01-22

Accept (Poster)